# Unsupervised Vision-Language Grammar Induction with Shared Structure Modeling

**Bo Wan[1], Wenjuan Han[2]\*, Zilong Zheng[2], Tinne Tuytelaars[1]**
1. Department of Electrical Engineering, KU Leuven;
2. Beijing Institute for General Artificial Intelligence, Beijing, China
{bwan;Tinne.Tuytelaars}@esat.kuleuven.be;
{hanwenjuan;zlzheng}@bigai.ai

## Abstract

We introduce a new task, unsupervised vision-language (VL) grammar induction. Given an image-caption pair, the goal is to extract a shared hierarchical structure for both image and language simultaneously. We argue that such structured output, grounded in both modalities, is a clear step towards the high-level understanding of multimodal information. Besides challenges existing in conventional visually grounded grammar induction tasks, VL grammar induction requires a model to capture contextual semantics and perform a fine-grained alignment. To address these challenges, we propose a novel method, CLIORA, which constructs a shared vision-language constituency tree structure with context-dependent semantics for all constituents in different levels of the tree. It computes a matching score between each constituent and image region, trained via contrastive learning. It integrates two levels of fusion, namely at feature-level and at score-level, so as to allow fine-grained alignment. We introduce a new evaluation metric: Critical Concept Recall Rate (CCRR) to explicitly evaluate VL grammar induction, and show a 2.6% improvement over a strong baseline on Flickr30k Entities. We also evaluate our model via two derived tasks, *i.e.*, language grammar induction and phrase grounding, and improve over the state-of-the-art for both.

## 1 Introduction

Humans are amazing at extracting knowledge efficiently from our complicated and multimodal world, leveraging both redundant and complementary information from visual, acoustic, or tactile cues. Investigating into such behavior, neuroimaging and neuroanatomical studies suggested that specific brain regions are dispatched to support the convergence of auditory and visual word comprehension (Calvert et al., 1997; Campbell, 2008; Keitel et al., 2020). For example, auditory regions are involved in lip reading by receiving signals from visual cortices (Bourguignon et al., 2020). These observations imply a mysterious "shared world" when perceiving multimodal signals, functioning as a centralized processor for understanding fused information. In contrast, such phenomena have not been revealed in modern state-of-the-art VL models, most of which process visual and language signals in two separate streams and fuse the results only in the final stage (Ma et al., 2015; You et al., 2018; Shi et al., 2019; Kojima et al., 2020; Zhao & Titov, 2020b).

In this work, we dive into the "shared world" for vision-language (VL) representations and introduce a new challenge – unsupervised VL grammar induction – aiming at extracting the shared hierarchical structure for both vision and language simultaneously. As a brief introduction, conventional grammar induction (Figure 1(a)), specifically constituency grammar induction, captures syntactic information in the form of constituency trees, which provide extra interpretability to downstream tasks, *e.g.*, semantic role labeling (Strubell et al., 2018), sentence completion (Zhang et al., 2016) and word representation (Kuncoro et al., 2020). It is commonly formulated as a self-contained system that relies solely on language corpora (Kim et al., 2019a; Drozdov et al., 2019; 2020; Shen et al., 2018; 2019). On top of this, Shi et al. (2019) proposes visually-grounded grammar induction, focusing on enhancing language grammar induction performance by leveraging additional visual information. Similar benefits of multi-modality on grammar induction have also been demonstrated

---

*Corresponding author. Code is available at https://github.com/bobwan1995/cliora.git

by Zhao & Titov (2020b); Zhang et al. (2021); Hong et al. (2021). These works, however, fail to consider a unified VL structure, nor have they demonstrated impact on visual understanding.

Different from prior arts, unsupervised VL grammar induction aims to construct a shared constituency structure at a fine-grained level for both the input image and the corresponding language caption; see Figure 1(b). It requires capabilities of structured prediction for a single-modality (language constituency structure) along with fine-grained alignment with heterogeneous modalities, while only having access to associated image-caption pairs for training (no human-generated ground truth) . Besides the general

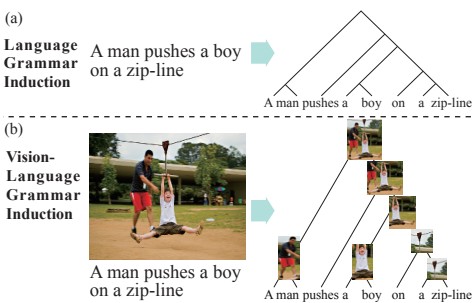

Figure 1: Task illustration of (a) conventional grammar induction for natural language and (b) VL grammar induction.

challenge of the unsupervised setting existing in conventional visually-grounded grammar induction tasks, we highlight two main challenges specific to our proposed new task:

1. *Context-dependent semantic representation learning.* The non-terminal symbol of a conventional constituency structure is a category label from a limited set (Hopcroft et al., 2001). (1) Such limitation leads to tractable learning but limited expressive power as it lacks semantics from words. In particular, sharing the same representation in the tree structure leads to semantic ambiguities. For example, two NP nodes in the tree represent two different phrases, but may have the same embedding. (2) Apart from lacking semantics from words, it also lacks rich contextual encoding of the phrases. Simply associating each symbol with specific words or contexts, as done in Zhu et al. (2020), would lead to memory explosion. (3) Besides textual data as context, exploiting the visual data as context information also remains a challenge.
2. *Fine-grained vision-language alignment for all levels of the hierarchical structure.* Instead of fusing information of the entire image into the phrase representations, as done in visually-grounded grammar induction (Shi et al., 2019; Zhao & Titov, 2020b), VL grammar induction requires each phrase in the structure to align with a specific Region of Interest (RoI) in the image. Such fine-grained alignment enables the model to have a thorough understanding of the image (Anderson et al., 2018; Zheng et al., 2019). However, the fine-grained alignment is difficult to deal with because the feature sets generated from different modalities are different in nature.

To address these challenges, we propose a potential approach, namely **C**ontrastive **L**anguage-**I**mage inside-**O**utside **R**ecursive **A**utoencoder (CLIORA). It leverages the previous success of DIORA (Drozdov et al., 2019) on context-dependent grammar induction for language and extends it in a multimodal scenario. Specifically, as sketched in Figure 2, it first extracts features from both modalities, then incorporates the inside-outside algorithm to compute the constituents and construct the constituency structure. Already at this stage, we combine the two modalities, by recursively having the language span embeddings attend to the visual features. We refer to this as *feature-level fusion.* This makes the phrases aware of the visual context, effectively exploiting the visual context as well as the textual semantics as context information, addressing the first challenge. On top of that, we compute a matching score between each constituent and image region. This score is used to promote the cross-modal fine-grained correspondence, leveraging the supervisory signal of the image-caption pairs via a contrastive learning strategy. Here, we further fuse the two modalities, by weighting the cross-modal matching score with the constituent's score given by the induced grammar. We refer to this as *score-level fusion.* This ensures fine-grained alignment in every level of the tree structure, addressing the second challenge.

In summary, our contributions are three-fold: (i) A new challenging task – unsupervised VL grammar induction for better cross-modal understanding, and a new metric to evaluate it; (ii) A novel method, CLIORA, to build the VL structure by context encoding with a multi-level fusion strategy; (iii) New state-of-the-art performance on MSCOCO and Flickr30k Entities benchmarks.

## 2 TASK DEFINITION

**VL Structure Formulation**    We introduce the shared VL constituency tree structure (VL structure) in Figure 1 to represent the shared semantics for cross-modality. In detail, given a sentence $\mathbf{x} =$

$\{x_1, x_2, ..., x_n\}$ with $n$ words and an associated image $I$, the VL structure $\mathbf{y}$, is formed in a phrase-structure tree similar to Chomsky Normal Form (CNF) (Chomsky, 1959), where each non-terminal node in the tree will have exactly two children. Formally, a VL structure $\mathbf{y}$ is a set of constituents[1] $\{(c_{i,j}, b_{i,j})\}$ that forms a tree structure of $\mathbf{x}$. Each non-terminal node of the tree contains a language span (said informally, phrase) $c_{i,j}$ corresponding to a sequence of words $\{x_i, x_{i+1}, ..., x_j\}$, and an aligned box region $b_{i,j} \in \mathbb{R}^4$ in the image $I$. $c_{i,j}$ is said to be grounded to $b_{i,j}$.

Different from the usual linguistic setting in context-free grammars (Hopcroft et al., 2001), the features of our non-terminal nodes contain rich context-dependent semantics instead of a category label. This results in a structure with powerful expressive ability, but simultaneously a high computational complexity if using conventional approaches (*e.g.*, Kim et al. (2019a)) to model it, as claimed in Yang et al. (2021); Han et al. (2017). On the image side, the structure provides explainable scene understanding. Different from the flat form of the structure typically used in phrase grounding (Wang et al., 2020a), each node in the hierarchical structure is associated with an image region. Different from scene parsing (Zhao et al., 2017), regions of different nodes are allowed to overlap.

**Task Formulation**   In unsupervised VL grammar induction, the goal is to induce phrase-structure grammars from only image-caption pairs without tree structure annotations nor phrase-region correspondence annotations for training[2]. Formally, we aim to learn a model $\mathcal{M}$, which takes an image $I$ and a language description $\mathbf{x}$ as input and predicts the VL structure $\mathbf{y}$, *i.e.*, $\mathbf{y} = \mathcal{M}(I, \mathbf{x})$. Note that different from Wang et al. (2020a) who predict the corresponding regions for a given set of noun phrases, noun phrases in VL grammar induction are unknown and all spans in the VL structure are aligned to  corresponding regions in the image.

**Evaluation Metrics**   Due to lacking annotations of VL structure, we indirectly assess our model by two derived tasks from each modality's perspective, *i.e.* language grammar induction and phrase grounding. Furthermore, we propose a new evaluation metric and conduct a frontal evaluation for the VL structure we obtained.

*Lateral Evaluation* For language grammar induction, we use two widely-used metrics: the averaged corpus-level F1 and averaged sentence-level F1 numbers along with the unbiased standard deviations following Zhao & Titov (2020a). For visual grounding, we report the grounding accuracy. We consider a noun phrase correctly grounded if its predicted bounding box has at least 0.5 IoU (Intersection over Union) with the ground-truth location. The grounding accuracy (ACC) is the fraction of correctly grounded noun phrases (from a given set of noun phrases).

*Frontal Evaluation* We propose a new metric, critical concept recall rate (CCRR), to explicitly evaluate the VL structure. A critical concept is a noun phrase found in visual grounding annotations. We say a critical concept is recalled when it is retrieved in the parsed constituency tree structure and correctly grounded in the image. CCRR is the recall rate of all the critical concepts.

## 3    CONTRASTIVE LANGUAGE-IMAGE INSIDE-OUTSIDE RECURSIVE AUTOENCODER

In this section, we design a novel VL grammar induction method, CLIORA , to construct a shared constituency structure for paired vision-language inputs. We start by briefly introducing our basis model DIORA (Drozdov et al., 2019) in Section 3.1. Then we present in detail our proposed CLIORA from modeling in Section 3.2, through inference in Section 3.3 to objective and learning in Section 3.4.

### 3.1    BACKGROUND

**Detailed Formulation of VL Structure**   Following Lafferty (2000); Drozdov et al. (2020), we adopt an indexing scheme for the constituency VL structure, and use a two-dimensional $n \times n$ chart

---

[1]In a usual grammar induction setting, the constituent (span) corresponds to a sequence of words. Here we reuse the conventional name but expand it to a pair of language span and aligned visual region.

[2]We do use a pre-trained object detector to obtain box regions, as is common in the phrase grounding literature. For a fully unsupervised setting, this could be replaced by a generic object proposal method (*e.g.*, Uijlings et al. (2013)), combined with features trained with self-supervision (Jaiswal et al., 2021).

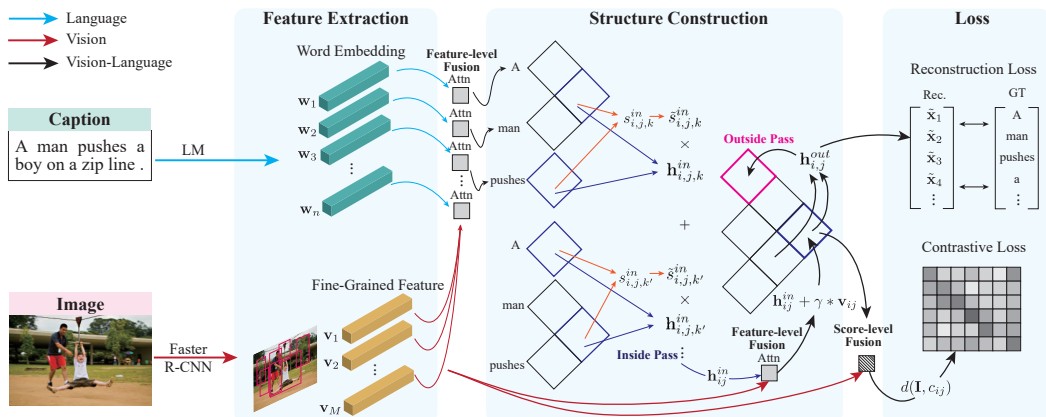

Figure 2: Diagram of CLIORA. The Feature Extraction module prepares the language and image features. The feature-level fusion enhances the language feature with visual cues. The Structure Construction module recursively computes the VL representation for constituents with feature-level fusion. Finally, the score-level fusion promotes the cross-modal fine-grained correspondence, leveraging the supervisory signal of the image-caption pairs.

$\mathbf{T}$ storing the intermediate representations while computing the spans of the VL structure. Cell $(i, j)$ in the chart contains all scores and vectors of span $c_{i,j}$. For each $1 \leqslant i < k < j \leqslant n$, span $(i, j)$ can be decomposed in spans $(i, k)$ and $(k + 1, j)$. Each span $c_{i,j}$ in the tree is associated with an inside score $s_{ij}^{in}$ and inside vector $\mathbf{h}_{ij}^{in} \in \mathbb{R}^D$ computed from bottom up (inside pass in next paragraph); as well as an outside score $s_{ij}^{out}$ and outside vector $\mathbf{h}_{ij}^{out} \in \mathbb{R}^D$ from top down (outside pass), with $D$ the feature dimension. The inside vector captures information of the inner content of the span, while the inside score assesses to what extent the span forms a phrase with complete semantics. Likewise, the outside score $s_{ij}^{out}$ and outside vector $\mathbf{h}_{ij}^{out}$ represent the contextual cues not in span $c_{i,j}$.

**DIORA** Deep Inside-Outside Recursive Autoencoder (DIORA) (Drozdov et al., 2019) aims to induce the language grammar and produce the constituency tree parser in an encoder-decoder framework. Different from context-free grammars, DIORA mitigates the strong context-freeness assumption by computing each span's representation and spans' composition possibility dependent on the context. DIORA operates like a masked language model since it models the context of a missing word and then reconstructs this missing word using the context as clue.

Formally, DIORA encodes the input sentence $\mathbf{x}$ in the shape of a constituency tree. Since the ground-truth tree structure is not given, all possible valid trees are considered simultaneously, with weights, using dynamic programming similar to the inside-outside algorithm (Baker, 1979). In the bottom-up inside pass, the encoder recursively runs an inside pass through all spans and computes the inside vector $\mathbf{h}_{i,j}^{in}$ and inside score $s_{i,j}^{in}$ for each constituent $c_{i,j}$. The combined constituent is obtained by weighted summation of all possible pairs split by $k \in [i, j)$. Similarly, the decoder performs a top-down outside pass, recursively computing the outside score $s_{i,j}^{out}$ and outside vector $\mathbf{h}_{i,j}^{out}$ with $k \in (1, i - 1] \cup [j + 1, n)$. In this way, the bottom-most vectors in the outside pass $\mathbf{h}_{i,i}^{out}$ encode the context of the entire sentence $\mathbf{x}$ except for the $i$-th word. During inference, the predicted tree with the maximum inside scores is obtained with the CKY algorithm (Kasami, 1966; Younger, 1967).

## 3.2 MODELING

Figure 2 illustrates the overall workflow, including visual/textual feature extraction, feature-level fusion, structure construction, score-level fusion, and the loss function module. The whole fusion process can be classified in feature-level (combining features vectors from different modalities) and score-level (combining the scores).

**Feature Extraction** For visual feature extraction, previous works on visually-grounded grammar induction often use only the global image feature (*e.g.*, global output feature in Shi et al. (2019); Zhao & Titov (2020b); Zhang et al. (2021)) as additional information. The detailed information from the image at region level, which is helpful in building the fine-grained VL correspondence, is ignored. Instead, we explore different aspects of visual features. In particular, we use an external object detector to generate a set of object proposals $\mathcal{O} = \{o_m\}_{m=1}^M$, where $o_m \in \mathbb{R}^4$ denotes an

image region (RoI). Similar to Wang et al. (2020b), for each $o_m$, we compute its conv-features and project it to a vector $\mathbf{v}_m \in \mathbb{R}^D$ with the same dimension $D$ as the span features. For textual features, following Drozdov et al. (2019), we use a pretrained word embedding $\mathbf{w}_i \in \mathbb{R}^D$ to initialize all the words $x_i$ in sentence $\mathbf{x}$. For more details c.f. Experiments Section.

**Feature-level Fusion**   The feature-level fusion strategy fuses the visual and language features, at different stages in our pipeline [3]: at the very start using the word embeddings $\mathbf{w}_i$, as well as during the recursive structure construction (described below), using the inside vectors $\mathbf{h}_{i,j}^{in}$. Denote $\mathbf{v}_{i,j}$ as the visual representation corresponding to span $c_{i,j}$. This is obtained by adopting an attention mechanism on the object features $\{\mathbf{v}_m\}_{m=1}^M$. In detail, we first compute the attention scores between $\mathbf{h}_{i,j}^{in}$ and each $\mathbf{v}_m$, then normalize the scores and obtain $\mathbf{v}_{i,j}$ by weighted summation over all object features. Finally, we enhance the span features $\mathbf{h}_{i,j}^{in}$ by fusing the visual information $\mathbf{v}_{i,j}$.

$$\mathbf{Att}_{i,j,m} = \mathrm{Soft}\max_m \{(\mathbf{v}_m)^{\mathrm{T}} \mathbf{h}_{i,j}^{in}\}; \qquad \mathbf{v}_{i,j} = \sum_m \mathbf{Att}_{i,j,m} \cdot \mathbf{v}_m; \qquad \hat{\mathbf{h}}_{i,j}^{in} = norm(\mathbf{h}_{i,j}^{in} + \gamma \cdot \mathbf{v}_{i,j})$$

$$(1)$$

where $\gamma$ is a weight value for fusion and $norm$ is L2 normalization.

**Structure Construction**   Inspired by DIORA, we use an autoencoder model to integrate the visual information and employ the inside pass and outside pass to fill in the chart $\mathbf{T}$. During the inside pass, for the leaf nodes, the scores $s_{i,i}^{in}$ are set to 0 and the span features $\mathbf{h}_{i,i}^{in}$ are initialized with the normalized word embeddings $norm(\mathbf{w}_i)$. During the outside pass, for the root node, the outside score $s_{1,n}^{out}$ is set to 0 and the language feature $\mathbf{h}_{1,n}^{out}$ is initialized randomly independent of $\mathbf{x}$. Then the inside-outside algorithm is employed to recursively calculate the scores $s_{i,j}^{in}, s_{i,j}^{out}$ and vectors $\mathbf{h}_{i,j}^{in}, \mathbf{h}_{i,j}^{out}$.

_Inside Pass_   The encoder is a bottom-up flow running an inside pass through all spans in the input sentence $\mathbf{x}$. It computes $\mathbf{h}_{i,j}^{in}$ and $s_{i,j}^{in}$ for each span $c_{i,j}$ to fill up each cell $(i,j)$ in the chart $\mathbf{T}$ as shown in Figure 2. We start by computing items for each decomposition $(i,k)$ and $(k+1,j)$.

$$\mathbf{h}_{i,j,k}^{in} = f(\hat{\mathbf{h}}_{i,k}^{in}, \hat{\mathbf{h}}_{k+1,j}^{in}) \qquad\qquad s_{i,j,k}^{in} = (\hat{\mathbf{h}}_{i,k}^{in})^{\mathrm{T}} \mathbf{W}(\hat{\mathbf{h}}_{k+1,j}^{in}) + s_{i,k}^{in} + s_{k+1,j}^{in} \qquad (2)$$

where $f$ is a composition function to merge the two spans (we use two linear layers with ReLU as activation function) and $\mathbf{W}$ are trainable parameters to compute the composition scores. $h_{i,j}^{in}$ is obtained by weighted summation over all possible pairs with the normalized $\tilde{s}_{i,j,k}^{in} = \mathrm{Soft}\max_k \{s_{i,j,k}^{in}\}$:

$$\mathbf{h}_{i,j}^{in} = \sum_k \mathbf{h}_{i,j,k}^{in} \cdot \tilde{s}_{i,j,k}^{in} \qquad\qquad s_{i,j}^{in} = \sum_k s_{i,j,k}^{in} \cdot \tilde{s}_{i,j,k}^{in} \qquad (3)$$

Then we fuse the visual information into $\mathbf{h}_{i,j}^{in}$ to obtain $\hat{\mathbf{h}}_{i,j}^{in}$ using Equation 1. $\hat{\mathbf{h}}_{i,j}^{in}$ in turn acts as a sub-span to recursively produce bigger spans, _e.g._, $\mathbf{h}_{i,j+1}^{in}$, $\mathbf{h}_{i-1,j+1}^{in}$ etc., until $\mathbf{h}_{1,n}^{in}$.

_Outside Pass_   The decoder performs a top-down outside pass, computing the outside scores $s_{i,j}^{out}$ and the outside vectors $\mathbf{h}_{i,j}^{out}$ by aggregating the inside representation and outside representation with the outside algorithm. $k \in (1, i-1] \cup [j+1, n)$. Take $k > j$ as an example[4]:

$$\mathbf{h}_{i,j,k}^{out} = f(\mathbf{h}_{i,k}^{out}, \hat{\mathbf{h}}_{j+1,k}^{in}) \qquad\qquad s_{i,j,k}^{out} = (\mathbf{h}_{i,k}^{out})^{\mathrm{T}} \mathbf{W}(\hat{\mathbf{h}}_{j+1,k}^{in}) + s_{i,k}^{out} + s_{j+1,k}^{in} \qquad (4)$$

We use the same composition function $f$ and score weight $\mathbf{W}$ as the inside pass. The computation of the outside score $s_{i,j}^{out}$ and outside feature representation $\mathbf{h}_{i,j}^{out}$ is similar to the inside pass with $\tilde{s}_{i,j,k}^{out} = \mathrm{Soft}\max_k \{s_{i,j,k}^{out}\}$:

$$\mathbf{h}_{i,j}^{out} = \sum_k \mathbf{h}_{i,j,k}^{out} \cdot \tilde{s}_{i,j,k}^{out} \qquad\qquad s_{i,j}^{out} = \sum_k s_{i,j,k}^{out} \cdot \tilde{s}_{i,j,k}^{out} \qquad (5)$$

In this way, the bottom-most vectors in the outside pass $\mathbf{h}_{i,i}^{out}$ encode the context of entire sentence $\mathbf{x}$ except for the $i$-th word.

---

[3] We explain it here for the latter case only. For the word embeddings, the procedure is basically the same.

[4] If $k < j$, the procedure is similar. Please refer to the Appendix D for more details.

**Score-level Fusion**  Finally, we fuse the span scores with visual similarity scores to obtain the input for our contrastive learning (detailed in Section 3.4). For this, we multiply the span score $q(c_{i,j}, \mathbf{x})$ (calculated using inside scores and outside scores, see below) with a similarity score $sim(I, c_{i,j})$, indicating the semantic similarity between span $c_{i,j}$ and the image $I$. This results in a distance between image $I$ and span $c_{i,j}$, that incorporates the VL structure information: $d(I, c_{i,j}) = sim(I, c_{i,j}) \times q(c_{i,j}, \mathbf{x})$.

*Similarity Score*  Following Wang et al. (2020b), we first compute the similarity score $a_{i,j,m}$ between each span $c_{i,j}$ and each image region $o_m$, then we select the image region with the highest similarity score, and use that score as similarity for the entire image:

$$a_{i,j,m} = \mathbf{v}_m^\mathrm{T}(\hat{\mathbf{h}}_{i,j}^{in} + \mathbf{h}_{i,j}^{out}) \qquad sim(I, c_{i,j}) = \max_m \{a_{i,j,m}\} \qquad (6)$$

Note how $a_{i,j,m}$ is calculated using both inside and outside feature vectors.

Similar to $sim(I, c_{i,j})$, which computes similarity at the span-level, we can also compute similarity scores at word level based on the original word embeddings: $a_{i,m}^w = \mathbf{v}_m^\mathrm{T}\mathbf{w}_i$. Analogically, $sim^w(I, x_i) = \max_m\{a_{i,m}^w\}$ then represents the similarity between a word and the entire image.

*Span Score*  $q(c_{i,j}, \mathbf{x})$ reflects how likely the span $c_{i,j}$ exists given $\mathbf{x}$. In conventional PCFG (Michael, 2011), since each span in the sentence is assigned to a non-terminal symbol, the marginal of the span $c_{i,j}$ is the summation over all non-terminal symbols. In CLIORA, we do not work with discrete symbols for the spans. Instead, we define $q(c_{i,j}, \mathbf{x}) = s_{i,j}^{in} \cdot s_{i,j}^{out}/s_{1,n}^{in}$. This is inspired by PCFG (please refer to the Appendix E for more details).

## 3.3  INFERENCE

In the inference stage, the language tree structure $\{c_{i,j}^*\}$ is predicted with the maximum inside scores using the CKY algorithm (Drozdov et al., 2019) on chart $\mathbf{T}$, and then the VL structure $\mathbf{y}^* = \{(c_{i,j}^*, b_{i,j}^*)\}$ is obtained by aligning each span $c_{i,j}^*$ to an image region $b_{i,j}^*$.

When a meaningful phrase (ground-truth constituent) $p = \{x_k : k \in [i, j]\}$ is given (e.g., for the visual grounding task), we predict its image region $\{o_{m_{i,j}^p}\}$ with a voting mechanism. We use $a_{k,m} = a_{k,m}^w + a_{k,k,m}$ (where $a_{k,m}^w$ and $a_{k,k,m}$ are defined in the Similarity Score paragraph above) as the matching score to select the most representative word $x_{k*}$ in $p$ with the maximum $a_{k,m}$, and take the box region $o_{m_{i,j}^p}$ based on the $x_{k*}$ with maximum matching score as $p$'s grounding result:

$$k^* = \arg \max_{i \leqslant k \leqslant j} \max_m a_{k,m} \qquad m_{i,j}^p = \arg \max_m a_{k*,m} \qquad (7)$$

When the meaningful phrase is not given, we can still predict the grounding region $o_{m_{i,j}^c}$ of an arbitrary span $c_{i,j}$. In that case, apart from $a_{k*,m}$ in Equation 7, we consider the RoI-span matching scores $a_{i,j,m}$ (defined in Similarity Score paragraph) to predict the aligned box $o_{m_{i,j}^c}$. This allows to assess how meaningful this span is given the visual information. The box region of $c_{i,j}$ is then:

$$m_{i,j}^c = \arg \max_m (a_{k*,m} + a_{i,j,m}) \qquad (8)$$

In Experiments Section, we adopt the first case in "Lateral Evaluation on Weakly-supervised Visual Grounding". In "Frontal Evaluation on VL Grammar Induction", we use the second case and obtain the grounding region $b_{i,j}^* = o_{m_{i,j}^c}$ for each predicted span $c_{i,j}^*$, since we do not have given ground-truth phrases as supervision.

## 3.4  OBJECTIVE AND LEARNING

**Loss for Structure Construction**  As mentioned in Structure Construction module, the bottom-most vectors in the outside pass $\mathbf{h}_{i,i}^{out}$ encode the context of the entire sentence $\mathbf{x}$ except for the $i$-th word. With the hypothesis that visual cues help to predict the missing words, CLIORA takes a self-supervised blank-filling objective as follows:

$$\mathcal{L}_{rec} = -\frac{1}{n} \sum_{x_i \in \mathbf{x}} \log P(x_i | \mathbf{h}_{i,i}^{out}) \qquad (9)$$

**Loss for Contrastive Learning**    We design our contrastive learning strategy based on maximizing the matching score between paired RoI-span elements instead of a coarse Image-span matching Zhao & Titov (2020a) or RoI-sentence matching (Wang et al., 2020b). Behind this design, our motivation is that compared with a whole sentence, a short span is more likely to find a corresponding RoI that can construct a strong negative pair to enhance contrastive learning. The same holds for the RoI compared with the whole image. Particularly, for each span $c_{i,j}$ in a caption sentence, all the other images except for the corresponding image $I$ in the current batch are negative examples. We can define the following $l_{span}(I, c_{i,j})$ and $l_{word}(I, x_i)$ in the batch:

$$l_{span}(I, c_{i,j}) = [d(I, c'_{i,j}) - d(I, c_{i,j}) + \epsilon]_+ + [d(I', c_{i,j}) - d(I, c_{i,j}) + \epsilon]_+ \quad (10)$$

$$l_{word}(I, x_i) = -\log \frac{e^{sim^w(I, x_i)}}{\sum_{\hat{I} \in batch} e^{sim^w(\hat{I}, x_i)}} \quad (11)$$

where $[\cdot]_+ = \max(0, \cdot)$, $\epsilon$ is the positive margin, and variables with $'$ are negative examples in a batch. A small $l_{span}$ means an aligned span-image pair that is closer than any  unaligned ones in a batch. Then we obtain the contrastive learning loss:

$$\mathcal{L}_{cl} = E_{p(\mathbf{y}|\mathbf{x})} \sum_{c_{i,j} \in \mathbf{y}} l_{span}(I, c_{i,j}) + \sum_{x_i \in \mathbf{x}} l_{word}(I, x_i) = \sum_{c_{i,j} \in \mathbf{x}} l_{span}(I, c_{i,j}) + \sum_{x_i \in \mathbf{x}} l_{word}(I, x_i)$$
$$(12)$$

where $\mathbf{y}$ is a valid tree for the sentence $\mathbf{x}$. Finally, the total loss is defined as:

$$\mathcal{L}_{\text{CLIORA}} = \lambda \cdot \mathcal{L}_{rec} + (1 - \lambda) \cdot \mathcal{L}_{cl} \quad (13)$$

where $\lambda$ is a hyper-parameter representing the trade-off between the two losses.

## 4    EXPERIMENTS

### 4.1    EXPERIMENTAL SETTING

**Datasets**    We evaluate our method on the Flickr30k Entities (Plummer et al., 2017) and MSCOCO (Lin et al., 2014) datasets. Flickr30K Entities contains 29783 images for training, 1,000 images for validation and 1,000 for test. We use the same split of MSCOCO as Zhao & Titov (2020b), which contains 82,783 training images, 1,000 validation images, and 1,000 test images. Each image is associated with 5 sentence descriptions for both Flickr30k Entities and MSCOCO datasets. Following Shi et al. (2019); Zhao & Titov (2020b), we filter words using the occurring frequency in the training set and build a vocabulary with size 10,000 for both Flickr30K Entities and MSCOCO. All sentences are converted to lowercase, discarding non-alphanumeric characters.

**Preprocessing**    Following MAF (Wang et al., 2020b), for an input image, we use an external object detector, Faster R-CNN (Ren et al., 2015), to generate object proposals and extract their visual features. Following Wang et al. (2020b), for each proposal, we use RoI-Align (He et al., 2017) and global average pooling to compute its conv-feature and embed

|  | **MSCOCO** | **Flickr30k** |
|---|---|---|
| *Only Language* | | |
| Left branch | 15.1 | 13.4 |
| Right branch | 51.0 | 1.7 |
| Random | 24.2±0.3 | 15.4±0.2 |
| C-PCFG* | 53.6±4.7 | 25.7±2.6 |
| DIORA† | 53.4±0.6 | 46.4±0.9 |
| DIORA | 58.0±0.7 | 54.3±2.0 |
| *Visually-Grounded* | | |
| VG-NSL | 50.4±0.3 | - |
| VG-NSL+HI | 53.3±0.2 | - |
| VC-PCFG* | 59.3±8.2 | 26.3±2.1 |
| Vision-Language (VL) | | |
| CLIORA† | 56.2±0.7 | 53.1±1.8 |
| CLIORA | 60.8±0.8 | 56.6±1.7 |

Table 1:    Grammar induction for language. Corpus-level F1 on the Test Datasets. ∗ means we re-implement the results on Flickr30k dataset as Zhao & Titov (2020b). † indicates we use randomly initialized word embedding.

it to a vector. We follow Drozdov et al. (2019) and Wang et al. (2020a) and use ELMo (Peters et al., 2018), and Glove embedding (Pennington et al., 2014) for MSCOCO and Flickr30k Entities, respectively. For the ground-truth structure of the captions, we follow Shi et al. (2019) and Zhao & Titov (2020a) to use predictions produced by Benepar (Kitaev & Klein, 2018).

**Setting**    For all experiments, we report the average score along with the unbiased standard deviations on four runs with different random seeds. We load DIORA as an initialization for CLIORA . Other detailed hyper-parameters are provided in Appendix F.

## 4.2 QUANTITATIVE RESULTS

**Lateral Evaluation on Language Grammar Induction** As shown in Table 1, we compare our approach with two classes of strong baselines: language grammar induction and visually grounded grammar induction. We use the corpus F1 as the metric. Our model outperforms the previous SOTA approaches that use only language information and those incorporating visual cues. In particular, CLIORA outperforms VC-PCFG by 1.5% corpus F1 score on MSCOCO dataset and 30.3% corpus F1 score on Flickr30k dataset.

Notably, PCFG-based models (C-PCFG and VC-PCFG) perform poorly on Flickr30k, while they work well on MSCOCO. Meanwhile, two classic baselines "Right branching" and "Left branching" perform very differently on MSCOCO and Flickr30k. We posit these phenomena are due to the different data distributions of MSCOCO and Flickr30k (See Figure 11 in Appendix). In contrast, CLIORA displays a more even performance across different corpora than PCFG-based models, which reveals its robustness across corpora.

**Lateral Evaluation on Weakly-supervised Visual Grounding** We evaluate our approach on weakly-supervised visual grounding task and compare CLIORA with current SOTA method MAF[5]. MAF only considers the ROI-word matching score $a_{i,m}^w$ as in Equation 7, while CLIORA leverages this ROI-word matching score as well as the context-aware RoI-span matching score $a_{i,i,m}$ in score-level fusion strategy. Not surprisingly, we observe a significant improvement by 2.1% ACC on Flickr30k, as shown in Table 2.

|  | ACC |
|---|---|
| MAF* | 50.4±0.1 |
| CLIORA | 52.5±0.7 |

Table 2: ACC on Flickr30k test set.

|  | CCRR |
|---|---|
| Baseline | 47.0±0.4 |
| CLIORA | 49.6±0.9 |

Table 3: CCRR on Flickr30k test set.

**Frontal Evaluation on VL Grammar Induction** In addition to the indirect lateral evaluations described above, we also directly assess the VL structure we obtained on the frontal evaluation metric CCRR. Our baseline adopts DIORA for grammar induction and MAF for span grounding (as described in Equation 7). CLIORA incorporates the cross-modal matching from the original word-region similarities and hierarchical RoI-span correspondence. Table 3 shows that our jointly learned VL structure outperforms the baseline with a large margin (2.6% on CCRR).

## 4.3 ABLATION STUDY ON FUSION STRATEGY

In this section we study the effectiveness of our two fusion strategies on language grammar induction: the feature-level fusion (FLF) to enhance feature representation and score-level fusion (SLF) to build structural cross-modal correspondence and provide regularization on meaningful spans. We leave out $l_{word}$ because the empirical study shows that it hardly affects the F1 score. As show in Table 4, FLF, which augments the language representation with vi-

| FLF | SLF | C-F1 |
|---|---|---|
| - | - | 54.3±2.0 |
| ✓ | - | 56.1±2.1 |
| - | ✓ | 55.9±2.4 |
| ✓ | ✓ | **56.6**±1.7 |

Table 4: Results on Flickr30k test set. FLF: Feature-Level Fusion. SLF: Score-Level Fusion.

sual context cues, is capable of improving the constituency parsing result with 1.8% on Corpus-F1. In addition, SLF takes advantage of RoI-span matching signal to supervise the learning of VL structure and benefits the grammar induction result with 1.6% on Corpus-F1. When simultaneously combining FLF and SLF, we observe a further improvement on the final result.

## 4.4 ANALYSIS ON SPAN LENGTHS AND LABELS

In details, we show the recall rate on six frequent constituent labels on Flickr30k Dev set, as well as Corpus F1 (CF1) and Sentence F1 (SF1) in Figure 3. Our model exhibits a general boost across constituent labels. Specifically, it is most accurate on ADJPs and ADVPs and works fairly well on PPs. We contribute this improvement to the benefits of using contextual semantics.

---

[5]MAF* indicates we use the same feature extractor and learning strategy as MAF. As the ground-truth noun phrase is only given in the inference stage, we use the corresponding region of the most representative word as the prediction of the phrase, which results in a lower number than the original paper.

Next, we analyze model performance on Flickr30k Dev set for constituents of different lengths in Figure 4. Surprisingly, CLIORA enhanced by image generally its non-visual version (*i.e.*, DIORA) on not only short phrases but also longer phrases. As claimed in VC-PCFG, visual information are beneficial for short spans but performs poorly on longer constituents($\geqslant 5$). Our model goes a step further in multi-modal fusion, proving that multi-modal fusion is still promising.

Figure 3: Recall on different constituent label.

Figure 4: Recall on different constituent length.

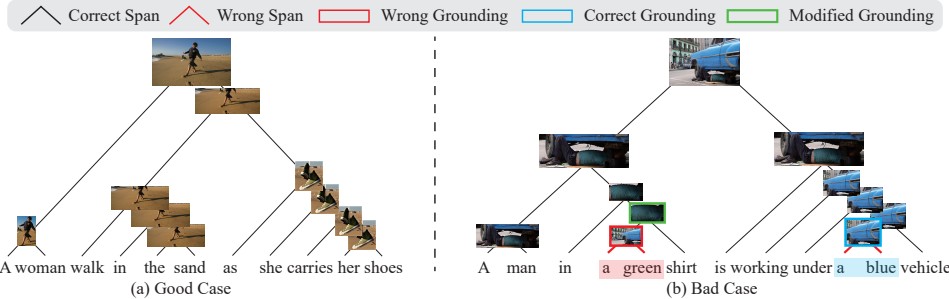

Figure 5: Case Study. RoIs and spans without special marks are predicted correctly.

# 5 LIMITATION AND FUTURE WORK

Although inspiring results have been achieved by CLIORA (See Figure 5(a) and Appendix H for success examples), the unsupervised VL grammar induction is far from satisfaction. Figure 5(b) demonstrates a typical failure case that wrongly predicts a span "a green" that should be "green shirt" in the ground-truth tree. This erroneous span is wrongly grounded to a large region containing a "green building". Interestingly, after combining more contexts and constituent a bigger span "a green shirt", the grounding result is modified to a correct region (green rectangle). This failure case reveals the importance of appropriately modeling contextual semantics, which highlights the major challenge of our proposed task and is calling for future research for better contextual models.

Moving forward, what is the best way of modeling such shared structure for VL grammar induction? A promising extension could be to explore the visual structure to regularize the shared VL grammar. It is worth noting that the visual images also contains enriched spatial and semantic structures (Si & Zhu, 2011; Zhao et al., 2017) which could be aligned with the constituency trees. Leveraging such structures may also be beneficial in producing a more meaningful shared structure.

Going back to the motivation of our work, how do humans model and process multimodal information with such shared space? This work provides a potential answer with respect to grammar induction and phrase grounding. Nevertheless, the debate between using dense vectors and symbolic structures in human's cognitive computational models has never been stopped (Tang et al., 2019). This mystery also leaves us a wide space to explore other potential explanations in modeling human's multimodal "shared world".

ACKNOWLEDGEMENT

We acknowledge funding from Flemish Government under the Onderzoeksprogramma Artificiele Intelligentie (AI) Vlaanderen programme. This work has also been supported by the ERC project 101021347 KeepOnLearning. We thank Yanpeng Zhao for helpful suggestions on this work.

ETHICS STATEMENT

Hereby, we consciously assure that our study is original work which has not been previously published elsewhere, and is not currently being considered for publication elsewhere. Our study doesn't have any threats to health, safety, personal security, and privacy. We do not have ethics risks as mentioned in the author guidelines.

REPRODUCIBILITY STATEMENT

All the codes, processed data, and the trained model in this paper are publicly released at https://github.com/bobwan1995/cliora.git. We use two public datasets: MSCOCO and Flickr30k Entities. We use the same split of MSCOCO dataset as VC-PCFG (Zhao & Titov, 2020b) and use their ground-truth tree structure annotation. We generate the ground-truth tree structure as VC-PCFG with Kitaev & Klein (2018) for Flickr30k Entities dataset. We re-implement the experiment results of CPCFG and VC-PCFG on Flickr30k Entities dataset in Table 1 with the same codes as VC-PCFG (Zhao & Titov, 2020b).

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

## A  RELATED WORK

**Grammar Induction for Language**   Grammar Induction for Language, especially unsupervised constituent parsing, is a fundamental task in Natural Language Processing that aims to capture syntactic information in sentences in the form of a phrase-structure tree. The constituency structure shows the process of analyzing the sentences by breaking down it into constituents. It finds its applications in semantic role labeling (Strubell et al., 2018) and word representation (Kuncoro et al., 2020) and many other tasks. While grammar induction has a long history in computational linguistics, previous studies are mostly limited to language domain with unannotated language corpora,

*e.g.*, Michael (2011); Kim et al. (2019a); Drozdov et al. (2019; 2020); Shen et al. (2019; 2018); Kim et al. (2019b). Recently, visually-grounded grammar induction, using visual images as perceptual experience of language, obtains increasing attention. Shi et al. (2019) first propose a visually grounded neural syntax learner, and Kojima et al. (2020) find such model is potentially biased towards concrete noun phrases. Then, more works (Zhao & Titov, 2020b; Zhang et al., 2021; Hong et al., 2021) further exploit visual semantics derived from images to improve continuous vector representations of language. Although these works get superior results on grammar induction, they don't consider building a unified vision and language structure.

**Weakly-supervised Visual Grounding**   Weakly-supervised Visual Grounding aims to build the fine-grained correspondence between language phrases and image regions with only image-caption pairs as supervision. Current works (Rohrbach et al., 2016; Chen et al., 2018; Gupta et al., 2020; Wang et al., 2021; Liu et al., 2021; Wang et al., 2020b) typically start with generating a set of region proposals from the input image by adopting an off-the-shelf object detector, and then formulate the grounding task as a weakly-supervised matching problem between the phrases and the proposals. GroundR (Rohrbach et al., 2016) built the cross-modal matching by reconstructing the input phrases with an attention mechanism on the visual features of the proposals. KAC-Net (Chen et al., 2018) took a similar pipeline but further utilized object categories as additional constraints and exploited visual consistency for supervision. Besides, some approaches (Gupta et al., 2020; Wang et al., 2021) exploited knowledge distillation from the language models and visual models pre-trained on an external corpus. RIR (Liu et al., 2021) explored the language structure from an external language parser that provided additional relation constrain on the noun phrases (Liu et al., 2020), and adopts a self-taught regression strategy to refine the box locations. MAF (Wang et al., 2020b) used a simple but effective contrastive learning strategy for visual representation learning (Kiros et al., 2014) and weakly-supervised visual grounding. VLGrammar (Hong et al., 2021) represents an alignment between the constituents of an image structure and language structure. Due to the separate image grammar, segmentation parts should be provided in advance which hinders its application in a broader domain.

## B   VL STRUCTURE FORMULATION

The VL structure is shown in Figure 6 (left). A shared vision-language constituency structure can be decoupled into a language constituency tree for grammar induction (Figure 6 (bottom right)) and grounding results for visual grounding (Figure 6 (top right)). From the visual angle, the structure provides a hierarchy scene understanding.

During this natural decoupling process, no training or extra parameters are needed. It reveals that building this VL structure requires the capability of structured prediction for single-modality along with fine-grained alignment with heterogeneous modalities. Due to this reason, this task is challenging. Nevertheless, building this VL structure benefits each separate task and obtains superior performance without using any extra annotations.

## C   TASK FORMULATION

Unsupervised VL grammar induction aims to induce a constituency grammar for both the caption and the image that the caption describes simultaneously. The VL structure $\mathbf{y}$ given the caption $\mathbf{x}$ is based on the constituency relations and defined as a 3-tuple $\mathcal{G} = \{\mathcal{N}, \Sigma, \mathcal{R}\}$, where $\mathcal{N}$ is the non-terminal nodes; $\Sigma$ is a finite set of terminal nodes, namely words; $\mathcal{R}$ represents the production rules including two classes of rules:

- $A \rightarrow BC; \quad A, B, C \in \mathcal{N},$

- $A \rightarrow x; \quad A \in \mathcal{N}, \quad x \in \Sigma.$

Each node $A \in \mathcal{N}$ located in cell $(i, j)$ is associated with a inside score $s_{ij}^{in}$, outside vector $\mathbf{h}_{ij}^{in}$ for the inside pass and outside score $s_{ij}^{out}$, outside vector $\mathbf{h}_{ij}^{out}$ for the outside pass.

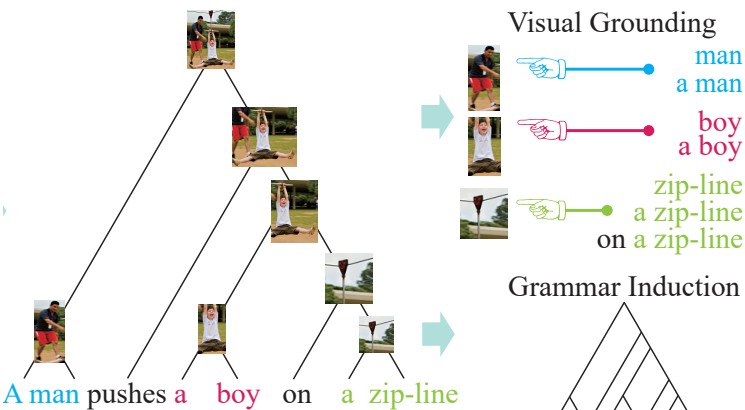

Figure 6: Illustration of the VL structure. Left: VL structure. Left→Right: decoupling of the VL structure. Bottom right: language constituency tree. Top right:grounding results. Different phrase are allowed to be aligned to the same region.

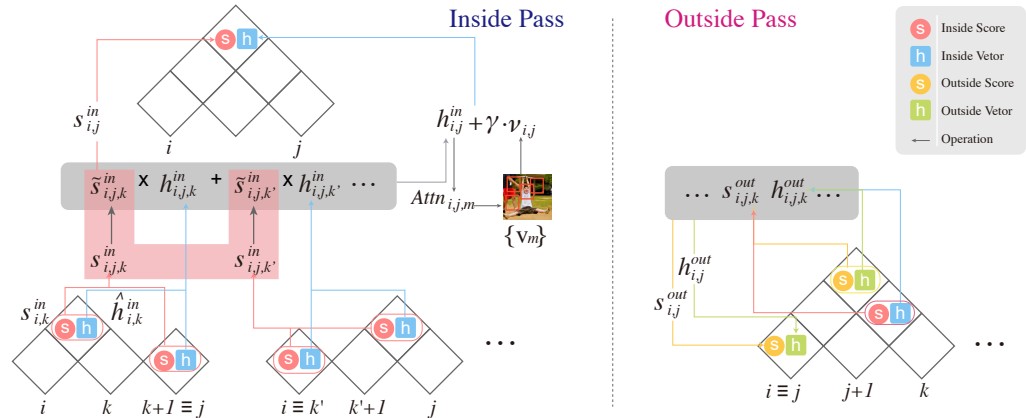

Figure 7: Detailed diagram of structure construction. The inside pass (left) and outside pass (right) are illustrated. Structure construction is the process of filling the chart $\mathbf{T}$. Rounds denote scores and rectangles denote vectors. Orange and blue shows variables in the inside pass. Yellow and green shows variables in the outside pass. Arrows refer to operations on data.

## D    STRUCTURE CONSTRUCTION

We follow Lafferty (2000); Drozdov et al. (2020) to use an indexing scheme for the constituency tree structure as shown in Figure 8. We use an autoencoder model to integrate the visual information

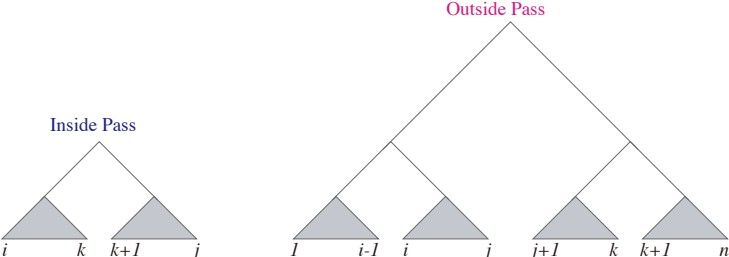

Figure 8: Inside pass and outside pass using the indexing scheme. For the inside pass (Left) two spans $c_{i,k}$ and $c_{k+1,j}$ is composed as a bigger span. For the outside pass (Right), representation of a target span $c_{i,j}$ is recursively computed from the inside representation of $c_{1,i-1}$ and outside representation of $c_{j+1,k}$ and $c_{k+1,n}$. $k$ can appear to the left or right of the target span $c_{i,k}$. Here we show $k$ on the right. If $k$ on the left, the indexing is adjusted.

and employ the inside pass and outside pass to fill in the $n \times n$ chart $\mathbf{T}$. For leaf nodes, $s_{i,i}^{in} = 0$ and $\hat{\mathbf{h}}_{i,i}^{in} = norm(\mathbf{w}_i) + \gamma \cdot I_{i,i}$ with a normalization layer. For Root node, the outside score $s_{1,n}^{out}$ is initialized as 0 and $\mathbf{h}_{1,n}^{out}$ is initialized randomly independent of $\mathbf{x}$. Then inside-outside algorithm is employed to recursively calculate the scores and vectors as shown in Figure 7.

_Inside Pass_    The encoder is a bottom-up flow by running an inside pass through all spans in the input sentence $\mathbf{x}$ and computes the inside vector $\mathbf{h}_{i,j}^{in}$ and inside score $s_{i,j}^{in}$ for each constituent $c_{i,j}$ to fill up each cell $(i,j)$ in the chart $\mathbf{T}$ as shown in Figure 2. More formally, we first compute items for each decomposition $(i,k)$ and $(k+1,j)$.

$$\mathbf{h}_{i,j,k}^{in} = f(\hat{\mathbf{h}}_{i,k}^{in}, \hat{\mathbf{h}}_{k+1,j}^{in}) \qquad s_{i,j,k}^{in} = (\hat{\mathbf{h}}_{i,k}^{in})^{\mathrm{T}} \mathbf{W}(\hat{\mathbf{h}}_{k+1,j}^{in}) + s_{i,k}^{in} + s_{k+1,j}^{in} \tag{14}$$

where the matrix $\mathbf{W}$ and the weights in $f$ are trainable parameters. Then the combined constituent is obtained by weighted summation of all possible pairs with the normalized $s_{i,j,k}^{in}$:

$$\tilde{s}_{i,j,k}^{in} = \underset{k}{\mathrm{Soft\,max}} \{s_{i,j,k}^{in}\} \qquad \mathbf{h}_{i,j}^{in} = \sum_k \mathbf{h}_{i,j,k}^{in} \cdot \tilde{s}_{i,j,k}^{in} \qquad s_{i,j}^{in} = \sum_k s_{i,j,k}^{in} \cdot \tilde{s}_{i,j,k}^{in} \tag{15}$$

_Outside Pass_    In contrast, the decoder performs a top-down outside pass, computing the outside score $s_{i,j}^{out}$ and the outside vector $\mathbf{h}_{i,j}^{out}$ by aggregating the inside representation and outside representation with the outside algorithm.:

$$\mathbf{h}_{i,j,k}^{out} = \begin{cases} f(\mathbf{h}_{i,k}^{out}, \mathbf{h}_{j+1,k}^{in}) & k > j \\ f(\mathbf{h}_{k,j}^{out}, \mathbf{h}_{k,i-1}^{in}) & k < i \end{cases} \tag{16}$$

$$s_{i,j,k}^{out} = \begin{cases} (\mathbf{h}_{i,k}^{out})^{\mathrm{T}} \mathbf{W}(\mathbf{h}_{j+1,k}^{in}) + s_{i,k}^{out} + s_{j+1,k}^{in} & k > j \\ (\mathbf{h}_{k,j}^{out})^{\mathrm{T}} \mathbf{W}(\mathbf{h}_{k,i-1}^{in}) + s_{k,j}^{out} + s_{k,i-1}^{in} & k < i \end{cases} \tag{17}$$

The computation of outside score $s_{i,j}^{out}$ and outside feature representation $\mathbf{h}_{i,j}^{out}$ is similar to the inside pass:

$$\tilde{s}_{i,j,k}^{out} = \underset{k}{\mathrm{Soft\,max}}\{s_{i,j,k}^{out}\} \qquad \mathbf{h}_{i,j}^{out} = \sum_k \mathbf{h}_{i,j,k}^{out} \cdot \tilde{s}_{i,j,k}^{out} \qquad s_{i,j}^{out} = \sum_k s_{i,j,k}^{out} \cdot \tilde{s}_{i,j,k}^{out} \tag{18}$$

In this way, bottom-most vectors in the outside pass $\mathbf{h}_{i,i}^{out}$ encodes the context of entire sentence $\mathbf{x}$ except for $i$-th word.

# E    SPAN SCORE

In PCFG, the marginal of the span $c_{i,j}$ is the conditional probability that is assigned after $\mathbf{x}$ is taken into account. Inspired by this, we define $q(c_{i,j}, \mathbf{x})$ to measure how likely the span $c_{i,j}$ exists given $\mathbf{x}$. Since each span in PCFG is assigned to a non-terminal symbol, the marginal of the span $c_{i,j}$ can be calculated using the summarization over all non-terminal symbols. However, due to the continuous representation for each span for CLIORA , we define $q(c_{i,j}, \mathbf{x})$ similar to PCFG. As shown in Figure 9, for PCFG since each span (namely, the string of words $\{\mathbf{x}_i...\mathbf{x}_j\}$ in the sentence $\mathbf{x} = \mathbf{x}_1...\mathbf{x}_n$ is assigned to a non-terminal symbol $A$. The marginal (ie., posterior) of the span $p(c_{i,j}|\mathbf{x})$ is the summarization over all non-terminal symbols $A \in \mathcal{N}$ ($\mathcal{N}$ is the set of all non-terminal symbols). The numerator of $p(c_{i,j,k}(B \to AC)|\mathbf{x})$ can be factorized as three parts: inside scores $a_{i,j}(A), a_{j+1,k}(C)$, outside score $b_{k,j}(B)$, production rule $p(B \to AC)$. The product of these three parts is finally divided by the partition function (the inside score of the whole sentence $a_{1,n}$.):

$$p(c_{i,j,k}(B \to AC)|\mathbf{x}) = \frac{a_{i,j}(A) \cdot a_{j+1,k}(C) \cdot p(B \to AC) \cdot b_{i,k}(B)}{a_{1,n}} \tag{19}$$

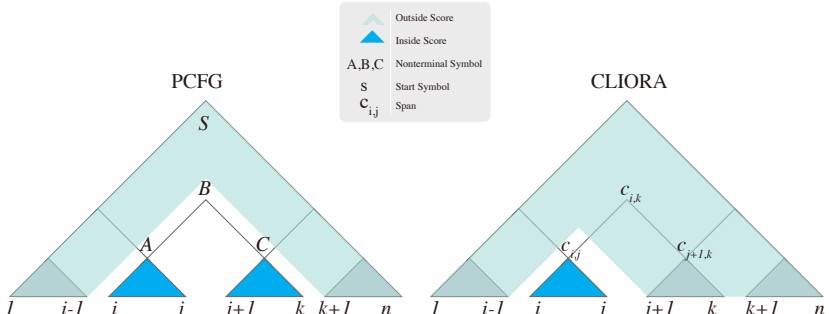

Figure 9: Calculation of the marginal and the span score. Right: marginal $p(c_{i,j}|\mathbf{x})$ of PCFG. The marginal of span $c_{i,j}$ is a fraction and the numerator consists of three parts: inside scores $a_{i,j}(A), a_{j+1,k}(C)$ (shaded in blue), outside score $b_{i,k}(B)$ (shaded in green), production rule $p(B \to AC)$. Left: span score $q(c_{i,j}, \mathbf{x})$ of CLIORA . The numerator of $q(c_{i,j}, \mathbf{x})$ can be factorized as two parts: inside scores $s_{i,j}^{in}$ (shaded in blue) and outside score $s_{i,j}^{out}$ (shaded in green). The product of these two parts is finally divided by $s_{1,n}^{in}$.

| | |
|---|---|
| $\lambda$ | 0.5 |
| $\gamma$ | 0.5 |
| # Epoch | 10 |
| Learning rate | $1e-5$ |
| Batch Size | 64 |

Table 5: Hyper-parameters for CLIORA .

However, the non-terminal symbol is not assigned to spans for CLIORA . Hence, the span score of span $c_{i,j}$ is defined as:

$$q(c_{i,j}, \mathbf{x}) = \frac{s_{i,j}^{in} \cdot s_{i,j}^{out}}{s_{1,n}^{in}} \qquad (20)$$

where $n$ is the length of the sentence $\mathbf{x}$. $i$ and $j$ represent the begin and end positions of the span in the sentence $\mathbf{x}$, respectively. The numerator of $q(c_{i,j}, \mathbf{x})$ can be factorized as two parts: inside scores $s_{i,j}^{in}$ and outside score $s_{i,j}^{out}$. The product of these two parts is finally divided by $s_{1,n}^{in}$.

## F  HYPER-PARAMETERS SETTING

The model selection is based on the loss on the development dataset. More concretely, we compute the average loss on the development set after each training epoch and select the model before the loss starts to increase. We list the main hyper-parameters in Table 5, and others are default as DIORA.

## G  DETAILED CASE STUDY

We visualize parse trees predicted by CLIORA in Figure 10. Left trees illustrate success cases and right trees are failure cases. For the top right tree of the failed cases, CLIORA wrongly predicts two spans "a green" and "a blue" that should be "green shirt" and "blue vehicle" in the ground-truth tree. The erroneous span "a green" is wrongly grounded to a green building. However, because there exists only one blue region, the erroneous span "a blue" is correctly grounded to the blue vehicle. Interestingly, after combining more contexts and constituent a bigger span "in a green shirt", the grounding result is modified to a correct region. This fail case reveals the importance of contextual semantics, which is an essential motivation of our task and model.

For the bottom right tree of the failed cases, CLIORA wrongly grounded the span "a boy and a girl" to the region referring to only the boy. We speculate that this error is due to the binary trait of the structure. "A boy" "and" and "a girl" should be combined simultaneously instead of wrongly combined the first two phrases then combine the last one. We will leave this for further study.

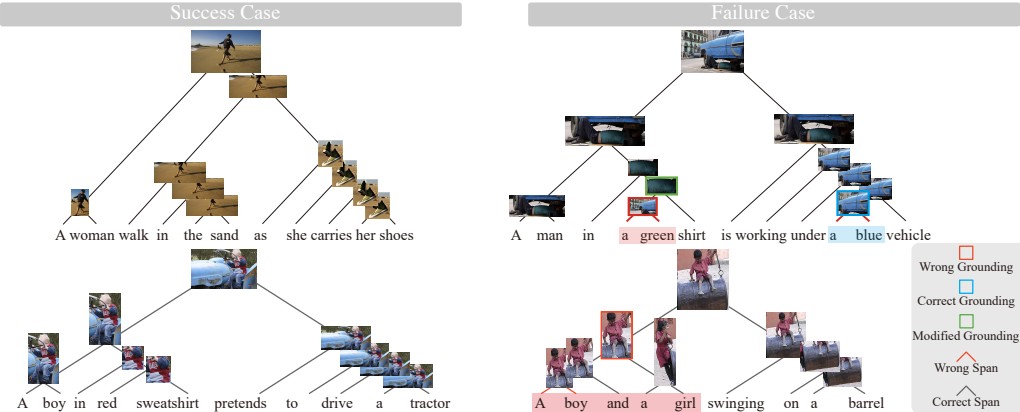

Figure 10: Case Study. RoIs without rectangles and spans without special marks are predicted correctly.

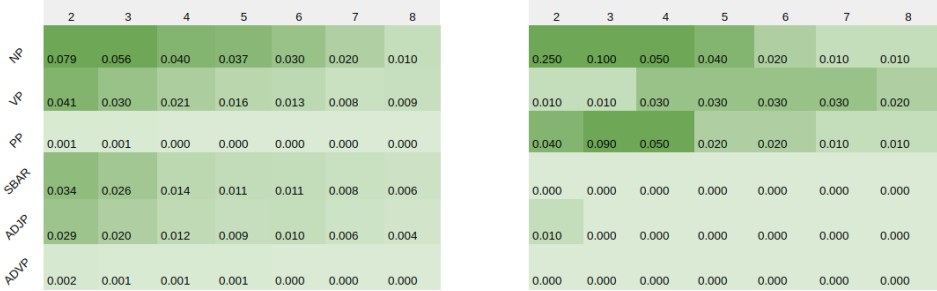

Figure 11: Label distribution over constituent length. The values in the cell denote frequencies of different constituent lengths and phrase types. Left: Flickr30k. Right: MSCOCO.

# H    ABLATION STUDY OF LANGUAGE AND IMAGE

We conduct an ablation study for contextual information and investigate the impact of image and text. After gradually removing the components, the model exhibit consistent degradation.

We use different forms of visual information in score-level fusion: global image information and fine-grained ROI information. If we remove the fine-grained information, which means the attention to the different ROIs are ignored and the average weight are used, the F1 score are even. It demonstrates that for score-level fusion, the global image information is enough. Additionally, to understand if the contribution of the visual compo-

|  | C-F1 |
|---|---|
| All | 54.3 |
| -Fine-grained ROI information | 54.2 |
| -Global image information | 51.8 |
| +Language Context | 48.0 |

Table 6: F1 on Flickr30k development dataset.

nent to the overall model is analogy with the language context, we add the representation of the whole sentence after removal of all visual information including fine-grained ROIs and the whole image. As for this replacement, the performance decreases, demonstrating that visual information can really helps in a compensation way, that is not restricted to providing contextual information like language.

# I    ROBUSTNESS ACROSS CORPORA

As shown in Table 1, different models perform very differently on MSCOCO and Flickr30k. PCFG-based models perform poorly on Flickr30k, while works well on MSCOCO. Two classic baselines "Right branching" and "Left branching" perform completely opposite on MSCOCO and Flickr30k. "Right branch" obtains a superior 51.0 F1 while "Left branch" only gets 15.1 on MSCOCO. "Right branch" performs worse than "Right branch" on Flickr30k. We posit this phenomenon is due to the diverse data distribution of MSCOCO and Flickr30k. To study this, we plot the detailed distribution over constituent length for different phrase types in Figure 11. Flickr30k contains various phrase types, *e.g.*, PP, SBAR and ADJP and longer constituents. In contrast, CLIORA displays a more even performance across different corpora than PCFG-based models, which reveals its robustness across corpora.

