# OpenReview forum: "Unsupervised Vision-Language Grammar Induction with Shared Structure Modeling"
_ICLR.cc/2022/Conference — ICLR 2022 Oral_

### Official Review · Reviewer_cvGw · 2021-11-01

**Correctness:** 3
**Technical Novelty And Significance:** 3
**Empirical Novelty And Significance:** 3
**Recommendation:** 8
**Confidence:** 3

**Main Review:**

Strengths:
The idea of jointly inducing structure in natural language and grounding the constituents with real-world images is intuitively correct.
The ablation study also shows that both feature-level fusion and score-level fusion (including the contrastive loss, if I understand correctly) helps in improving the parsing results.

Weakness:
1) The image features are only used for computing the inside pass. The image feature should contain information that can help predict the missing word, such that it could be used in the outside pass too. Selecting the best image region for predicting the missing word is also an intuitively correct way to build the vision-language alignment.
2) The compute of sim(I, c_ij) includes a max operator, this could lead to a biased gradient.
3) As the author mentioned in the discussion section, the model doesn't consider the latent hierarchical structure of the image. For example, the sentence describes the entire image, while each phrase describes part of the image.

**Summary Of The Paper:**

The paper proposed a new method CLIORA to do unsupervised parsing and vision-language grounding.
CLIORA is based on DIORA model.
But different from previous unsupervised parsing methods, CLIORA also induces alignment between constituents and image regions.
In order to train the model, the author introduces a contrastive loss.
Experiment results show that the proposed method outperforms baseline unsupervised parsing methods and it also induces meaningful alignment between image regions and constituents.

**Summary Of The Review:**

Overall, the proposed method is interesting and inspiring.
The idea should be interesting to both unsupervised parsing and multimodel communities.

---

> ### Author Response · Authors · 2021-11-18
> **Response to Reviewer cvGw**
>
> Thank you for acknowledging the novelty of our proposed method. All your concerns are addressed as follows:
>
> **Q: Image features**
>
> **A:** In the outside pass we aggregate the inside representation and outside representation with the outside algorithm (see Appendix D for the detailed description). As the inside representation contains visual cues, thus each outside representation is also aware of the visual information. Actually we tried to use the aggregated visual feature from the inside representation to reconstruct the original word, i.e., replace $P(x_i|h_{i,i}^{out})$ in Eq. 9 with $P(x_i|h_{i,i}^{out}, v_{i,i})$. However, we didn’t observe a performance gain with such modification. We suppose this might be because the outside representations already contain the visual information.
>
> --------------
>
> **Q: Max operator**
>
> **A:** We follow MAF (Wang et al., 2020b) and use the max operation for contrastive learning. The motivation of taking max operation is that visual grounding is remarkably unambiguous in the sense that only a tiny portion of image areas are assigned to a language phrase. Indeed the max operation will lead to a biased gradient. The empirical results show that the max operation works well in our method.
>
>
> --------------
>
> **Q: Hierarchical structure of the image**
>
> **A:** As we discussed in Section 5, the hierarchical visual structure is critical for joint VL grammar induction and we plan to explore it in future work.

---

> > ### Comment · Reviewer_cvGw · 2021-12-01
> > **Thanks**
> >
> > Thank you for your clarifications. My score remains as it was (positive). I look forward to future work on the hierarchical visual structure.

---

### Official Review · Reviewer_3QG1 · 2021-11-02

**Correctness:** 3
**Technical Novelty And Significance:** 3
**Empirical Novelty And Significance:** 3
**Recommendation:** 8
**Confidence:** 3

**Main Review:**

The topic of grammar induction has been there for a very long time in NLP and is a very fundamental topic.  The model was largely built based on an existing model for purely text-based grammar induction. The model essentially makes use of neural networks to learn good latent representations (using a reconstruction loss) where the latent representation is defined with neural networks which yield scores for constituents and vector representations of them. The approach adopts the classical inside-outside process for the computing of the scores.

The paper essentially investigates what might be the effective methods for integrating image information into text for improved grammar induction. The execution of the paper was quite good, and the results are convincing. However, I feel the overall model is essentially a way to use image information to regularize the grammar induction process. Little can be said about in what precise manner the image is actively contributing to the induction process. Indeed, the authors also acknowledged something along with what I thought in the final section. Nevertheless, I think it is an interesting piece that might inspire future research on multimodal processing (for image + language).

**Summary Of The Paper:**

This paper presents a new model for grammar induction for text, with help from the coupled images. The model was built on top of an existing unsupervised grammar induction model used for text without image information. The experimental results show the approach was effective. The work essentially demonstrates some effective ways of leveraging the additional image information for improving the grammar induction task. The paper also discussed some weaknesses of the approach and future work.

**Summary Of The Review:**

I think this is a reasonable piece, with good writing and a nice set of experiments. It would be helpful for future research in this domain.

---

> ### Author Response · Authors · 2021-11-18
> **Response to Reviewer 3QG1**
>
> Thank you for acknowledging the impact of our work.
>
> **Q: what precise manner the image is actively contributing to the induction process.**
>
> **A:** The main purpose of this work is to induce a shared VL grammar. In this work, we not only use the visual information to regularize the language grammar induction, but also transfer the language structure to the visual space to construct a shared VL structure by using the context-aware VL representations. To select the meaningful spans, we expect the fine-grained image information (i.e. the visual objects) to provide the visual-grounded information (Shi et al.  (2019); Zhao & Titov (2020b)). Besides, such visual objects also provide global context information to enhance the feature representation. The explanation behind such enhancement is out of scope for this work and we leave it to future research on explainable modeling.

---

### Official Review · Reviewer_hzHn · 2021-11-02

**Correctness:** 3
**Technical Novelty And Significance:** 3
**Empirical Novelty And Significance:** 3
**Recommendation:** 8
**Confidence:** 5

**Main Review:**


### Strengths

- As far as I know, this is the first paper that proposes joint visual-linguistic grammar induction in a real-world setting (in contrast to synthetic settings; Hong et al., 2021).

- The approach and the evaluation process are solid and make a lot of sense to me.

- The visually grounded parsing results are quite impressive.

### Weakness

- My major concern is about the model selection process and the potential unfair comparisons to existing work.
    - Model selection: If I understood correctly, for text parsing, the best models are selected w.r.t. to the parsing performance on a 1000-example dev set (Appendix F). \
This is an unrealistic setting (see https://aclanthology.org/2020.emnlp-main.614.pdf for discussions; in short, for any fancy unsupervised parsing model that uses a labeled development set, a supervised model trained on these development examples should be considered as a strong baseline) -- introducing unsupervised criteria for model selection is more important than our initial impression.

    - Unfair comparison: CLIORA, the model proposed in this paper, uses DIORA as initialization, which uses ELMo to initialize word embeddings and the PTB labeled development set for model selection. This means that CLIORA has seen far more text than other baselines (VG-NSL, C-PCFG, VC-PCFG, and so on) and human language learners. \
This issue also undermines the authors' arguments about potential links to how humans learn language. I expect either a CLIORA trained from scratch (without DIORA initialization) or weakened arguments about the relationship between the current CLIORA and human language learning.

- There seem to be some confusion on basic linguistic concepts, e.g., nonterminal vs. terminal symbols, and a few typos that affects smooth understanding (please see also detailed comments below).

### Other comments and questions

- Introduction: "These works, however, fail to consider a unified VL structure, nor have they demonstrated impact on visual understanding."  \
I don't think I necessarily agree with this statement, especially regarding Hong et al. (2021). Despite that there is a clear gap between their dataset and the real world settings, they are aligning the "visual grammars" to language grammars, yielding an arguably unified VL structure.

- Introduction: "The non-terminal symbol of a conventional constituency structure is a category label from a limited set (e.g., the set of part-of-speech (POS) tags) (Hopcroft et al., 2001). " \
Do you mean *terminal* symbols here? We usually refer to POS tags (to clarify, phrase tags are not POS tags) by preterminal or terminal (depending on whether the phrase-structure grammar is lexicalized, i.e., whether it's considering real words or just POS tags), and refer to the phrase nodes by nonterminal nodes/symbols (e.g., NP, PP). It seems that this is not a typo -- I have the same questions for the following task definition section on page 3.

- Task definition, evaluation metrics: if I understood correctly, CCRA requires some extra annotation of critical concepts -- how did you collect such annotations to determine which NPs are critical? \
(Very minor) based on the full name, CCRA should really be CCRR -- what does A stand for here?

- Section 3.2, feature extraction: the Yoon Kim et al. (2019b) paper is not relevant to image features at all -- did you mean Shi et al., (2019)?

- Table 1: what is the dagger after VGNSL-HI?

- Section 4.3: did you mean "augments" by "arguments"?

- Some more thoughts regarding motivation limitations: humans arguably learns how to parse concrete sentences first, and can then generalize to abstract domains that are not visually groundable. In this work, it seems that the model only works when both the text and image are available, as there is a need to infuse visual features into text spans. Do you have any thoughts on enabling a trained CLIORA model to parse pure text without grounding signals?

### Missing Reference

Kojima et al. [1] has strengthened the VG-NSL model by simplifying the architecture, and argued that such visually grounded models are potentially biased towards concrete noun phrases. However, the paper neither cited it nor discussed the relevant issues.

[1] https://aclanthology.org/2020.acl-main.234.pdf

There have been a lot of relevant work earlier than 2019 on visual-semantic embeddings or structured visual understanding with text. To name a few,

**Older work on structured image-text representations**

[2] https://openaccess.thecvf.com/content_iccv_2015/papers/Ma_Multimodal_Convolutional_Neural_ICCV_2015_paper.pdf

[3] https://openaccess.thecvf.com/content_cvpr_2018/papers/You_End-to-End_Convolutional_Semantic_CVPR_2018_paper.pdf

**Contrastive loss for visual-semantic embeddings**

[4] https://arxiv.org/pdf/1411.2539.pdf


### Minor Editing Comments
- I was confused about what CCRA is when reading the abstract -- would be good to include the full name and give an intuitive description of the metric.

- Yoon et al. $\rightarrow$ Kim et al.

- Shi et al. (2019) proposes $\rightarrow$ Shi et al. (2019) propose

- In my opinion, putting Section 3.4 before 3.3 would better streamline the paper.


**Summary Of The Paper:**

This paper introduces a task of joint visual-linguistic grammar induction from parallel image-text data, presents models and metrics for the task, and shows strong empirical results.


**Summary Of The Review:**

This paper introduces the task of joint visual-linguistic grammar induction, and presents models, metrics and empirical results on it. While I appreciate the impressive results, I am concerned about the unrealistic model selection process (comparing model outputs to a large set of ground-truth parse trees) and the unfair comparison (the proposed model has access to much more unlabeled text data than baselines).

---

> ### Author Response · Authors · 2021-11-18
> **Response to Reviewer hzHn**
>
> Thank you for the valuable suggestions and references. We hereby carefully address your concerns as follows:
>
> **Q: Model selection**
>
> **A:** Thanks for pointing this out. Model selection is a classic problem in an unsupervised setting due to the indirect evaluation w.r.t. the learning objective (see comments of Kyunghyun Cho in PRPN https://openreview.net/forum?id=rkgOLb-0W). Recent SOTA approaches, e.g. VC-PCFG and DIORA, have not mentioned if they select their checkpoint according to the annotations from the dev data.
>
> To evaluate our model’s robustness to different model selection approaches, we follow a similar strategy as in PRPN and use the loss on the dev set (without using the annotations in the dev set) for model selection. More concretely, we compute the average loss on the dev set after each training epoch and select the model before the loss starts to increase. With such a strategy, we observe minor performance drop in both the baseline (0.4%) and our method (0.6%) compared with using the annotations on the dev set to select the final model (The results are conducted on the Flickr30k Entities dataset, and the trends are similar on MSCOCO). However, the performance gain (2.3%) of our method is still obvious. We update all the experiment results with such a strategy in the revised version.
>
> ----------------------------------------
>
> **Q: Unfair comparison**
>
> **A:** Indeed, our model CLIORA is built upon DIORA which uses ELMo/Glove to initialize word embeddings and outperforms all the other works. To further address your concern, we remove the initialization of word embeddings in DIORA and build CLIORA upon it. Besides, we take the new strategy for model selection as mentioned above. The new experimental results are listed below and supplemented in the revision (Tab. 1). Impressively, when we start from such a weak baseline, our CLIORA obtains a large performance gain of 6.7% compared with DIORA and still outperforms VC-PCFG with 26.8% on the Flickr30k dataset. Such observation further validates our motivation that CLIORA can jointly learn vision and language structures.
>
>
> |        | **MSCOCO**  | **Flickr30k**
> |------ | ------|----------
> |DIORA† |  53.4±0.6 | 46.4±0.9
> |DIORA |   58.0±0.7 |  54.3±2.0
> |CLIORA†   | 56.2±0.7     |  53.1±1.8
> |CLIORA | 60.8±0.8   |   56.6±1.7
>
> († indicates we use randomly initialized word embedding.)
>
>
> ----------------------------------------
>
> **Q: Introduction (Discussion of “These works, however, fail to consider a unified VL structure”)**
>
> **A:** We respectfully disagree. We argue that the structure in Hong et al. (2021) is not “unified” in terms of the representation perspective. In their setting, they build a language and a vision constituency tree separately and match the nodes in the two trees. However, the structures of the two trees are not the same, as the number of terminals of the sentences (i.e. language words) and visual objects (i.e., object parts) is not the same. In our paper, we build a shared structure for vision and language.
>
> ----------------------------------------
>
> **Q: Introduction (Nonterminals Clarification)**
>
> **A:** Here, we mean the nonterminals. It’s the same for the task definition section on page 3. We have clarified this in the revision.
>
> ----------------------------------------
>
> **Q: Task definition & evaluation metrics**
>
> **A:** We define a critical concept as a meaningful phrase in the ground-truth constituency tree. In other words, “meaningful” phrases are phrases that can be grounded in the image. As there does not exist a dataset that fully labels all the critical concepts, we select a certain type of critical concept, the noun phrases, from the visual grounding annotations (i.e. Flickr30k Entities annotations). We think a critical concept is recalled when it is retrieved in the parsed constituency tree structure and simultaneously correctly grounded in the image. All the noun phrases are critical in the GT annotations.
>
> **CCRR:** Thanks for the suggestion. We have made changes accordingly.
>
> ----------------------------------------
>
> **Q: Enabling a trained CLIORA model to parse pure text**
>
> **A:** Our CLIORA adopts two fusion strategies to leverage the information from vision and language: feature-level and score-level fusion. Actually, if we only take score-level fusion, the visual cues only help the training of CLIORA and are not used in the inference stage, which means we take only pure text as input. As shown in the third row of Tab. 4, with only score-level fusion, we can still obtain 1.6% performance gain compared with the baseline.
>
> ----------------------------------------
>
> **Minor Comments**
>
> **A:** Thanks so much for the advice on the abstract and for pointing out the typos and the missing references.  All have been fixed, including:
> - CCRA $\rightarrow$ CCRR
> - Yoon Kim et al. (2019b) $\rightarrow$ Shi et al., (2019)
> - Section 4.3: arguments $\rightarrow$ augments
> - Remove the dagger after VG-NSL+HI

---

> > ### Comment · Reviewer_hzHn · 2021-11-18
> > **Happy to see that most of my comments are resolved, and further discussion**
> >
> > Thanks authors for your response! I appreciate that most of my comments have been addressed. I'm now inclined to raise my rating -- will update it after the discussion period concludes. Below are the points on which I am still not convinced:
> >
> > ---
> > **Unified Grammar**
> >
> > While I agree with the authors that this work is the first one proposing a "unified" vision-language grammar, if "unified" means that an image and parallel text have exactly the same parsing structure, I am not sure if it is an advantage (or disadvantage) to have a fully synchronous vision-language grammar.
> >
> > Consider an image that has several semantically equivalent captions with different syntactic structures, the image would have different parsings associated with different captions. This is very unnatural to me on the vision side, especially when linking to how humans acquire language: we arguably don't parse the visual scene in exactly the same way as how we parse text. The visual parses in this work are therefore less meaningful than text parses.
> >
> > To clarify, I do not mean that modeling the visual structures in the current way is bad: it provides ways to better understand which visual part each text span corresponds to, and, from the engineering perspective, it enables easier fusion with text features. However, the current paper and ablation study did not convince me that such the "unified" VL setting should be claimed as an advantage compared to existing work, instead of just one choice among many possibilities.
> >
> > ---
> > **Author Response**: "Recent SOTA approaches, e.g. VC-PCFG and DIORA, have not mentioned if they select their checkpoint according to the annotations from the dev data."
> >
> > **R #hzHn**: While I'm happy to see the authors have done experiments with (fully unsupervised) loss based model selection and have clarified in Appendix F, I would like to emphasize that existing work in a problematic setting, especially when it has been concerned for years, should never be an excuse for conducting additional work in the setting.
> >
> > DIORA does select the best model based on the annotated dev data (https://aclanthology.org/N19-1116.pdf; Appendix A.1 Early Stopping). At the meantime, I am slightly worried about the hyperparameter settings (Appendix F): DIORA uses the PTB dev set for choosing the best parameters, while the authors choose their tuned hyperparameters, and only do unsupervised model selection.
> > Ideally, the hyperparameters should also be tuned based on the current development loss.
> >
> > ---
> >
> > There are still some words that refer to part-of-speech tags as nonterminals (e.g., Section 3.2, Appendix E).
> > I'm looking forward to seeing these fixed.

---

> > > ### Author Response · Authors · 2021-11-21
> > > **Re: Happy to see that most of my comments are resolved, and further discussion**
> > >
> > > We appreciate your careful reading of our response and considering raising your rating. Based on your comments, we would like to further address your concerns as follows:
> > >
> > > **Unified Grammar:** “if it is an advantage (or disadvantage) to have a fully synchronous vision-language grammar.”
> > >
> > > **A:** To clarify, in this work, the “unification” is referred to as constructing a common VL representation for vision and language simultaneously. We focus on the “shared world” for VL representations. The motivation behind our interest is in the “shared world” (i.e., the synchronous vision-language structure to reveal the cross-modality semantic consistency) to enhance multi-modality understanding. Empirically, we have demonstrated that such a structured learning paradigm not only benefits the language grammar induction but also weakly-supervised visual grounding.
> > >
> > > We agree that “the image would have different parsings associated with different captions”. As we discussed in Section 5, the work, though demonstrates the concept of shared structure representation, still has the limitations of failing to leverage the visual structure information. By proposing this new structure representation, we hope to provide another perspective of multimodal structure modeling (compared with visually-aided ones) and call for future research to explore its potential (i.e. advantages / disadvantages).
> > >
> > > An interesting and potential point may be considering not only the “shared world” but also the “individual world”. Different modality has its own characteristics. Some previous works (Krishna et al., 2017) have claimed this opinion and build hierarchical structures for visual scenes (e.g., scene graphs) and natural languages (e.g., dependency/constituent trees), individually. Respecting the semantic consistency of the image-sentence pair as well as independent characteristics, a joint vision-language structure considering what is the shared structure (e.g., the shared VL structure we proposed in this work) and what is the individual structure for each modality is definitely a further work that is worth further studying.
> > >
> > >
> > > **Re: R #hzHn:** “...DIORA uses the PTB dev set for choosing the best parameters, while the authors choose their tuned hyperparameters, and only do unsupervised model selection... ”
> > >
> > > **A:** We have performed a couple of hyperparameter tuning experiments to justify our selection for the model’s initialization. To assess DIORA’s hyperparameter selection in an unsupervised way, we use the same criterion as our model selection approach, and tune DIORA’s hyperparameters based on the loss on the development dataset (without using the unannotated data). More concretely, we compute the average loss on the Flickr30k Entities development set for each hyperparameter value and select the hyperparameter with the smallest loss. Take the essential hyperparameter -- the learning rate as an example, we choose the learning rate from values of [1e-6, 1e-5, 1e-4]. When using a large learning rate (e.g., 1e-4), the average loss of selected modes is 17.8 and the C-F1 is 34.8±3.8. With learning rates 1e-5 and 1e-6, the average loss decreases to 4.7 and the C-F1 is stable at a value as reported. Therefore, for our model, we can adopt an unsupervised approach for both the hyperparameter tuning and the model selection.
> > >
> > >
> > > **Words that refer to part-of-speech tags as nonterminals**
> > >
> > > **A:** Thank you for pointing this out! We have had them fixed (including Section 3.2, Appendix E).

---

> > > > ### Comment · Reviewer_hzHn · 2021-11-21
> > > > **Thank you for your response.**
> > > >
> > > > Thanks for the clarification on your point about the unified structure, and for getting the results on unsupervised hyperparameter tuning.
> > > > I now confirm that the paper is valuable to be presented at ICLR -- I'll raise my rating to show clear recommendation towards acceptance.

---

> > > > > ### Author Response · Authors · 2021-11-21
> > > > > **Re: Thank you for your response.**
> > > > >
> > > > > Thank you so much for raising the score and providing detailed suggestions for the revision. We believe your recommendation would also benefit the development of the multimodal community.

---

### Author Response · Authors · 2021-11-18
**Response to All Reviewers**

We thank all reviewers for their time and valuable comments. In this work, we propose a new task VL grammar induction that aims to build a shared VL structure.  We appreciate reviewers acknowledge that:
- This work introduced “the first visual-linguistic Grammar Induction in a real-world setting” (Reviewer hzHn), which “is intuitively correct” (Reviewer cvGw), and “might inspire future research on explainable multimodal processing” (Reviewer 3QG1).
- Our proposed method CLIORA is “built based on an existing model” (Reviewer 3QG1) and is “interesting and inspiring” (Reviewer cvGw). The method and evaluation “are solid and make a lot of sense.” (Reviewer hzHn)
- The parse results are “impressive” (Reviewer hzHn) and “convincing” (Reviewer 3QG1).

We sincerely hope that this work would benefit the explainable multimodal community. We have made some changes to address the comments from the reviewers. We would really appreciate it if the reviewers could read our Rebuttal Revision.

---

### Public Comment · ~Bo_Wan1 · 2022-04-13
**Codes and data release.**

Codes and data are available at https://github.com/bobwan1995/cliora.

---

### Decision · Program_Chairs · 2022-01-20

**Decision:**

Accept (Oral)

**Comment:**

This paper proposes to perform unsupervised grammar induction over image-text pairs and used shared structure between the modalities to improve grammar induction on both sides. Authors find the paper clear, creative and interesting and recommend acceptance without hesitation.